# An Analysis of BERT FAQ Retrieval Models for COVID-19 Infobot

**Shuo Sun**
Johns Hopkins University
ssun32@jhu.edu

**João Sedoc**
Johns Hopkins University
jsedoc@jhu.edu

## Abstract

The outbreak of the COVID-19 pandemic has caused tremendous amounts of suffering and deaths around the world and greatly affected the lives of humanity. As the world sees more infected cases every day, the need and demand for **reliable and up-to-date** information on COVID-19 have never been higher. While recent pre-trained language models show successes on many other NLP tasks, we did not have COVID-19 related dataset to help us evaluate the performance of QA systems and infobots based on these models. After the creation of a COVID-19 question similarity dataset by public health experts from the Johns Hopkins Bloomberg School of Public Health (JHSPH), we create models sufficient for application. We also analyze the amount of supervised data required.

## 1 Introduction

The COVID-19 pandemic has undeniably affected the lives of almost everyone in every part of the world. Schools are closed, companies are shutting down permanently, and people are losing jobs due to the lack of consumer demands. While doctors, nurses, and many other essential workers are at the front-line battling the virus, many concerned citizens are at home, searching for the latest developments of the pandemics and keeping them up to date with the newest information and guidelines from organizations such as CDC and WHO. However, misinformation is rampant in social media () and even public officials e.g. ingestion of disinfectants or use of NSAIDs, such as aspirin and ibuprofen. This motivates the need to answer questions like "Should I ingest disinfectants to treat COVID-19?" and "Can I use Aspirin with COVID?" The desire for **reliable and up-to-date** information related to a pandemic has never been greater in this modern era. Consequently, NLP practitioners quickly ramp up QA systems that are designed to automatically answer COVID-19 related questions.

Traditional, QA systems can be categorized into generation-based methods (Serban et al., 2016; Xing et al., 2018) which synthesize answers using natural language generation techniques, and retrieval-based methods (Wu et al., 2018; Sakata et al., 2019), which retrieve the best answers from a list of given candidate answers. Given the existence of a vast amount of publicly available question-answer pairs from FAQ webpages maintained by organizations such as WHO[1] and CDC[2], most existing COVID-19 QA systems use retrieval-based methods. We can further classify the retrieval-based techniques into three subcategories:

**Rule-based** These QA systems follow a set of predefined rules (Frederking, 1981) when generating responses to human questions. The rules are usually curated manually and require constant updates as the COVID-19 situations evolve around the world. They are also prone to errors caused by the insufficiency of rules to cover different situations. For example, QA systems that look for the coexisting keywords "what" and "COVID-19" to generate responses for the question "What is COVID-19?" might also produce similar answers to "What is the incubation period of COVID-19?".

**Q-A Similarities** QA systems in this category compute similarity scores between input questions and candidate answers and then sort candidate answers base on the similarity scores. The question-answer pairs can be ranked with traditional Information Retrieval (IR) methods such as tf-idf (Salton and McGill, 1986) and BM25 (Robertson et al., 2009; Chen and Van Durme, 2017) or neural

---

[1]https://www.who.int/news-room/q-a-detail/q-a-coronaviruses

[2]https://www.cdc.gov/coronavirus/2019-ncov/faq.html

IR methods (Sasaki et al., 2018; McDonald et al., 2018). Recently, models based on pre-trained language models such as BERT (Devlin et al., 2019; MacAvaney et al., 2019; Reimers and Gurevych, 2019) have demonstrated strong performance on sentence similarity and retrieval tasks.

**Q-Q Similarities** QA systems in this category are similar to systems based on Q-A similarities, except that they calculate similarity scores between input questions and candidate questions instead of candidate answers. In other words, these QA systems retrieve and return the answers of candidate questions that are most similar to the input question.

In this work, we explore the feasibilities of using pre-trained language models to compute Q-A and Q-Q similarities for retrieval-based COVID-19 QA systems. To support our experiments, we created a preliminary COVID-19 question similarity dataset in collaboration with experts from the Johns Hopkins Bloomberg School of Public Health (JH-SPH). Evaluation results on our preliminary dataset suggest that although fine-tuned BERT-based models perform decently in terms of IR metrics, these models *do not perform at the precision levels justifiable for direct real-world applications*. Further, our experiments also suggest it is challenging to find threshold similarity scores that can balance the precision and recall for these models. We argue that high-precision systems are exceptionally important at this crucial moment since we do not want to serve irrelevant information to worried users, or worse, inadvertently disseminate false information. We further show that with some supervision from our dataset, the overall performance of these models improves significantly. To support further researches, we will publicly release our COVID-19 question similarity dataset soon.

## 2 Approaches

Figure 1 presents the system architecture of a typical baseline retrieval-based QA system. We first build a database of candidate question-answer pairs by scraping COVID-19 related frequently asked questions (FAQ) web pages from a list of carefully chosen data sources. A retrieval-based QA system ingests an input question and returns top-ranked candidate question-answer pairs from the dataset based on similarities between the input question and candidates in the database. At the time of submission for this paper, our database contains

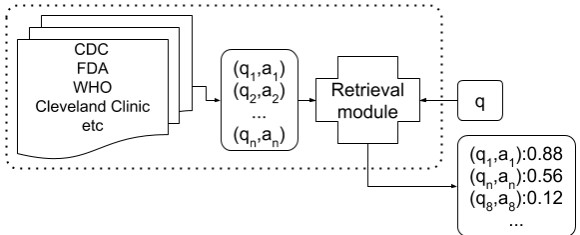

Figure 1: The system architecture of a retrieval-based COVID-19 QA system. FAQ webpages are scraped from reliable sources such as CDC, FDA, and WHO and pooled together into a database of question-answer pairs. A retrieval module ingests an input question and returns top-ranked candidate question-answer pairs from the database based on computed similarity metrics.

690 question-answer pairs extracted from 12 data sources. We will use this architecture for all experiments in this paper.

Since we want to examine the effectiveness of existing retrieval solutions, we experiment with two common used retrieval techniques:

**BM25** The BM25 model (Robertson et al., 2009) is a well-known ranking function commonly used in search engines. It is a bag-of-word model that calculates similarity scores between the terms in queries and the terms in documents. We adapt BM25 to the QA task by treating input questions as queries and the question-answer pairs as documents. We use Elasticsearch[3], which uses BM25 by default, as our backend retrieval system.

**BERT** This is a state-of-the-art pre-trained language model that performs well on many NLP tasks. BERT (Devlin et al., 2019) and its variants such as Roberta (Liu et al., 2019) haven been consistently producing top results on the SQUAD2.0 (Rajpurkar et al., 2018) leaderboard. Recently, Sakata et al. (2019) shows that BERT-based FAQ retrieval systems outperform baseline retrieval systems on benchmark IR datasets. In this paper, we experiment with BERT models in both unsupervised and supervised settings:

1. Under unsupervised setting, we use sentence transformers[4] (Reimers and Gurevych, 2019) to encode the input questions and candidate questions (or candidate answers) into semantically meaningful BERT sentence embeddings. The sentence transformers are BERT-based

---

[3]https://www.elastic.co/
[4]https://github.com/UKPLab/sentence-transformers

models that were fine-tuned on publicly available natural language inference (NLI) and semantic text similarity (STS) datasets. The sentence embeddings from these models are also aligned, meaning that the cosine similarities between sentence embeddings reflect their degrees of similarities. We can then calculate similarity scores between the input question and candidate questions (or candidate answers) by taking the cosine similarity between their sentence embeddings.

2. Under supervised setting, we further fine-tune the sentence transformers with examples from our COVID-19 question similarity dataset.

For every model, we run experiments in both *Q-Q mode* where we calculate similarity scores between input questions and candidate questions, and *Q-A mode* where we compute similarity scores between input questions and candidate answers. We report results in mean average precision (MAP) (Buckley and Voorhees, 2005) and normalized discounted cumulative gain (NDCG) (Järvelin and Kekäläinen, 2002).[5]

## 3 Dataset

Due to the subjective nature of evaluating QA systems and the lack of in-domain data related to Covid19, we are creating a new COVID-19 question similarity dataset in collaboration with experts from the Johns Hopkins Bloomberg School of Public Health (JHSPH). The annotation process can be summarized as follows:

1. We use a filtered subsample of user-generated questions from Qorona[6], a list of COVID-19 related questions collected using Google autocomplete API, and from COVID-19 related data collected by DialogueMD[7].

2. For each input question, we retrieve the top five question-answer pairs from a pool of candidate question-answer pairs[8] with the help of a BM25-based baseline QA retrieval system.

3. We engage public health experts to directly assess the relevance of the candidate question-answer pairs on a scale of 0–100.

4. For input questions with no retrieved relevant question-answer pairs, our annotators manually craft answers for those questions.

An example from our dateset is shown in figure 2.

At the time of submission of this paper, our preliminary dataset contains 6495 input questions with 32475 candidate question-answer pairs, covering a large variety of questions such as "Can COVID-19 be spread through surface-touching?" and "Can we use fabric masks to prevent the spread?". We reserve 1497 questions for the test set and use the other 4998 annotated instances for training.

## 4 Experimental Setup

We filter out instances with no relevant candidates and some instances with blank candidate answers. Our filtered benchmark test set contains 392 examples. We assign relevance labels of one to question-answer candidates with annotated scores $\geq 80$ and zero otherwise.

Ideally, for a given input question $\gamma$ and a list of candidate question-answer pairs $C = \{(q_1, a_1) \ldots (q_5, a_5)\}$, we want to learn a function f, such that $f(\gamma, (q_i, a_i)) > f(\gamma, (q_j, a_j)) \Leftrightarrow g(\gamma, (q_i, a_i)) > g(\gamma, (q_j, a_j))$ for $1 \leq i, j \leq 5$, where g is a function that returns the annotated relevance label. For our BM25 baseline, $f$ is modeled by the BM25 ranking function in Elasticsearch. For BERT-based models, $f$ returns the cosine similarity between the sentence embedding of an input question and the sentence embedding of a candidate question or candidate answer.

We conducted all experiments on an AWS instance with 8 cpus, 60GB of RAM and a 16GB Nvidia Tesla V100 GPU.

## 5 Results

Table 1 presents the results of various models on test set of our preliminary COVID-19 question similarity dataset. We highlight some of the findings here:

First, models that were fine-tuned on our annotated dataset significantly improve the performances on the COVID-19 question similarity test set. For example, NDCG@3 of a BERT retrieval model fine-tuned on NLI data improves from 0.544 to 0.626 and from 0.309 to 0.626 when we fine-tune that model on similarities between (input question, candidate question) pairs and (input question, candidate answer) pairs respectively. The surprise

---

[5]Both metrics can be calculated with the pytrec_eval tool (Van Gysel and de Rijke, 2018).

[6]https://github.com/allenai/Qorona

[7]https://github.com/dialoguemd/COVID-19

[8]We scraped FAQ webpages from reliable sources such as CDC, FDA, WHO and Cleveland Clinic.

**Question:** Can I go for a run Does running exercise compromise my immune system
**Candidate 1:** (We are currently on lockdown... can I go outside? Can I work out outside? Can I go for a run? Can I go for a walk?, ...) → **100**
**Candidate 2:** (Should I go to work if there is an outbreak in my community?, ...) → **0**
**Candidate 3:** (Can I take my child to the playground? → **0**, ...)
**Candidate 4:** (Can i go to the funeral of someone who died of COVID-19?, ...) → **0**
**Candidate 5:** (How can I and my family prepare for COVID-19?, ...) → **0**

Figure 2: An example from our COVID-19 question-answering dataset. For every input question, we retrieved five candidate question-answer pairs using a baseline BM25 retrieval system. Annotators were asked to carefully assign relevance scores between 0 – 100 to the candidates.

| Model | Fine-tune | Q-Q mode | | | | Q-A mode | | | |
|---|---|---|---|---|---|---|---|---|---|
| | | MAP | N@1 | N@3 | T/Q(s) | MAP | N@1 | N@3 | T/Q(s) |
| BM25* | N/A | 0.569 | 0.523 | 0.572 | – | 0.461 | 0.370 | 0.474 | – |
| **Unsupervised** | | | | | | | | | |
| BERT | NLI | 0.537 | 0.477 | 0.544 | 0.018 | 0.334 | 0.194 | 0.309 | 0.030 |
| Roberta | NLI | 0.529 | 0.464 | 0.535 | 0.048 | 0.337 | 0.194 | 0.315 | 0.066 |
| BERT | NLI→STSB | 0.504 | 0.426 | 0.511 | 0.018 | 0.386 | 0.225 | 0.413 | 0.030 |
| Roberta | NLI→STSB | 0.505 | 0.423 | 0.507 | 0.047 | 0.334 | 0.189 | 0.303 | 0.066 |
| CovidBERT | NLI | 0.533 | 0.462 | 0.544 | 0.018 | 0.318 | 0.176 | 0.277 | 0.031 |
| **Supervised – Trained on (input question, candidate question) pairs** | | | | | | | | | |
| BERT | None | 0.614 | 0.587 | 0.619 | 0.018 | 0.460 | 0.304 | 0.493 | 0.030 |
| BERT | NLI | 0.623 | 0.605 | 0.626 | 0.018 | 0.411 | 0.268 | 0.457 | 0.030 |
| CovidBERT | NLI | 0.617 | 0.592 | 0.622 | 0.018 | 0.474 | 0.321 | 0.509 | 0.032 |
| TwitterBERT | None | 0.621 | 0.600 | 0.624 | 0.018 | 0.396 | 0.270 | 0.398 | 0.030 |
| **Supervised – Trained on (input question, candidate answer) pairs** | | | | | | | | | |
| BERT | None | 0.605 | 0.577 | 0.611 | 0.017 | **0.620** | **0.600** | **0.626** | 0.030 |
| BERT | NLI | 0.605 | 0.579 | 0.611 | 0.018 | 0.615 | 0.587 | 0.623 | 0.029 |
| TwitterBERT | None | 0.597 | 0.566 | 0.603 | 0.017 | 0.579 | 0.548 | 0.580 | 0.030 |
| CovidBERT | NLI | 0.609 | 0.584 | 0.614 | 0.017 | 0.618 | 0.597 | 0.624 | 0.030 |
| **Supervised – Trained on both** | | | | | | | | | |
| BERT | None | 0.615 | 0.589 | 0.618 | 0.018 | 0.614 | 0.587 | 0.619 | 0.030 |
| BERT | NLI | **0.624** | **0.607** | **0.627** | 0.018 | 0.617 | 0.594 | 0.621 | 0.030 |
| TwitterBERT | None | 0.614 | 0.587 | 0.621 | 0.018 | 0.611 | 0.579 | 0.619 | 0.030 |
| CovidBERT | NLI | 0.614 | 0.584 | 0.621 | 0.018 | 0.612 | 0.584 | 0.618 | 0.030 |

Table 1: MAP and NDCG (cut off at top 1 and top 3 documents) of various retrieval models. Q-Q mode ranks candidates based on similarity scores between input questions and candidate questions, while Q-A mode ranks candidates based on similarity scores between input questions and candidate answers. T/Q is the average time (in second) taken to calculate similarity scores for each input question. All BERT models are based on BERT-base-cased and all Roberta models are fine-tuned on Roberta-large. CovidBERT was (continue) trained on AllenAI's CORD19 Dataset of scientific articles about coronaviruses. TwitterBERT was (continue) trained on tweets about coronavirus.

here is that models that were fine-tuned on only (input question, candidate question) pairs also significantly outperform unsupervised models when we evaluate those models in Q-A mode. For example, the NCCG@3 of the same BERT model improves from 0.309 to 0.493 when evaluated in Q-A mode. We hypothesize that some of the candidate questions are summaries of the candidate

answers and because of that, the sentence representations of the candidate questions might be close to the sentence representations of the candidate answers. Therefore, learning to align the vectors of input questions and candidate questions would also improve the alignment between the vectors of input questions and candidate answers.

Second, we observe that unsupervised models *perform significantly better in Q-Q mode than in Q-A mode.* For example, unsupervised models can perform at NDCG@3 of around 0.507 to 0.544 in Q-Q mode, but their performances drop significantly to around 0.266 to 0.309 in Q-A mode. This also applies to the supervised models trained on (input question, candidate question) pairs which perform at NDCG@3 of around 0.619 to 0.626 in Q-Q mode against 0.398 to 0.493 in Q-A mode. This is expected given the fact that those models were fine-tuned on short sentence pairs, which is different from the answers in our COVID-19 dataset that are significantly longer. In contrast, models that were fine-tuned on (input question, candidate answer) pairs or both (input question, candidate question) and (input question, candidate answer) pairs perform well in both Q-Q and Q-A modes.

Third, although Roberta outperforms BERT on many benchmark datasets (Rajpurkar et al., 2018), it does not seem to perform better than BERT on our benchmark COVID-19 test set. As we can see from the unsupervised section of table 1, BERT outperforms Roberta under almost all settings. Further, because Roberta models have significantly more parameters than BERT models, they take 2–3 times longer to compute sentence embeddings and cosine similarities for every batch of data. We exclude Roberta from further experiments and focus on BERT models for the remaining of this paper.

Last but not least, vanilla BM25 model using the default parameters from Elasticsearch outperforms all unsupervised BERT-based models in both Q-Q and Q-A modes. In contrast, it perform worse than the supervised models in Q-Q mode and Q-A mode.

In general, unsupervised BERT-based models perform decently well on our benchmark test set, performing at NDCG@1 of around 0.423 to 0.477, which means that these models can rank relevant candidates at the top one positions around 42.3% to 47.7% of the time.

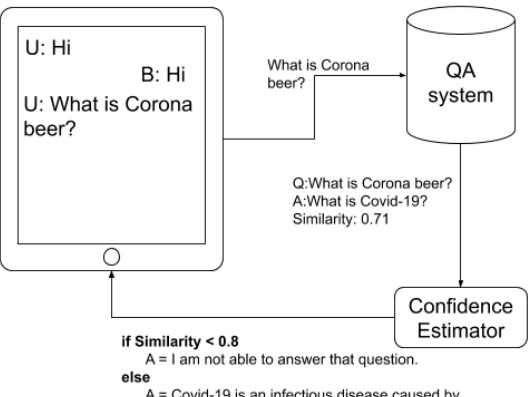

Figure 3: A COVID-19 QA system serving as the back-end system of a COVID-19 infobot. The QA system contains a database of question-answer pairs similar to the one seen in figure 1. As the system is not perfect, there are cases where the QA system returns incorrect results or cannot find valid answers in the database. An additional confidence estimator is needed to filter out bad results.

## 6 Applying QA to COVID-19 infobot

Unlike typical QA retrieval systems that are designed to show users lists of top-ranked candidate answers and let the users decide what are the best answers, infobot expects the QA system to return the most confident answer. In other words, an infobot should serve answers to input questions if and only if it is confident that the answers are correct. If not, the infobot should explain to users that it does not know how to answer the questions as seen in figure 3. We want to further emphasize the importance of precision in this setting since we do not want to provide irrelevant answers to users, or worse, give wrong advice to users.

Therefore, a confidence estimator is needed to filter out irrelevant or wrong answers. A commonly used approach in the NLP community is to set a threshold to the similarity scores. As seen in the example in figure 3, any candidate answer with similarity score of less than 0.8 will be rejected and replaced with "I am not able to answer that question".

To evaluate how well our retrieval systems do in a infobot-based environment, we measure the performances of our models in terms of precision, recall, and F1 at different threshold values. We collect results at 101 threshold values between 0.0 and 1.0, evenly spaced out at the interval of 0.01. For each threshold value, a candidate is considered correct, if the similarity score between the candidate

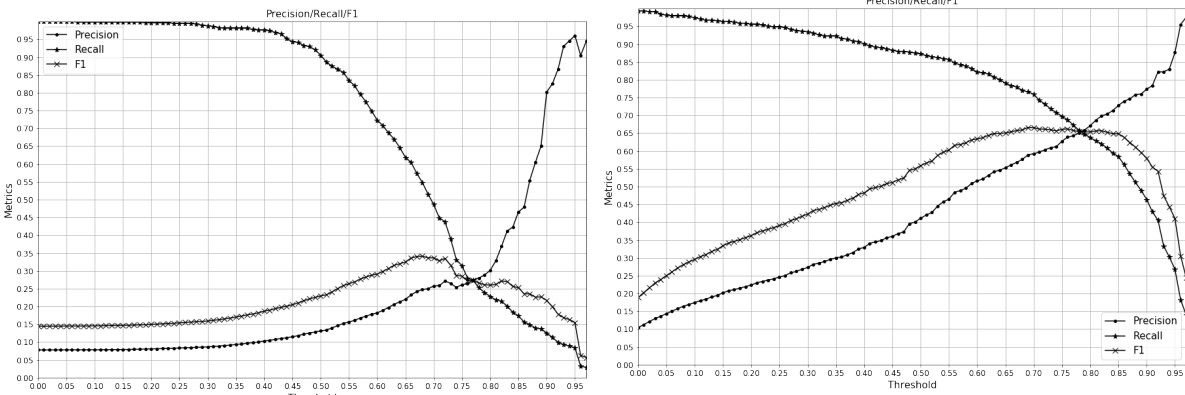

(a) Precision/Recall/F1 curves of an unsupervised BERT model. (b) Precision/Recall/F1 curves of the model from figure 4a fine-tuned on our COVID-19 dataset.

Figure 4: Precision/Recall/F1 curves of an unsupervised model versus a supervised model.

and the input question is greater than the threshold value. We gather all (input question, candidate) tuples from our COVID-19 question similarity test set and then convert them into true/false labels according to the threshold. We calculate the precision, recall, and F1 values between the predicted outputs and the actual relevance labels at all threshold values.

We show the precision, recall, and F1 curves of an unsupervised BERT-NLI model before and after it was fine-tuned on our annotated dataset. Both models were evaluated in Q-Q mode and we expect the trend is similar to other unsupervised and supervised models.

As seen in figure 4, the unsupervised model performs poorly at this task, achieving a maximum F1 score of less than 0.35,and the three metrics converge at a low value of around 0.27. In contrast, the situation is much better for the supervised model, where the best F1 score is more than 0.65, and all three metrics also converge at around 0.65. We hypothesize that the scales of cosine similarities from the unsupervised model are different for different sentences, therefore it is difficult to find a global threshold that works well for all sentences. In comparison, our annotated dataset optimizes those scales and makes it easier to find a reasonable threshold.

## 6.1 How much training data is actually needed?

Our results show that it is possible to improve the F1 from around 0.35 to 0.65 by fine-tuning those models with our annotated dataset. An interesting question then arises as to what percentage of train-

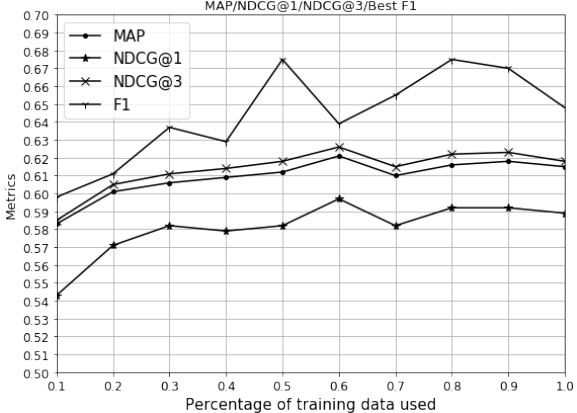

Figure 5: MAP/NDCG@1/NDCG@3/F1 against percentage of training data used.

ing data is needed to reach peak performance. To find out, we re-trained a vanilla BERT model on sub-samples of our training data. As seen in figure 5, with just 10% of the training data, the model achieves a good F1 of 0.598 and NDCG@3 of 0.543. However, it only manages to hit its peak F1 when trained on 50% of the training data and hit its peak MAP and NDCG when trained on 60% of the training data. Those percentages translate to 2499 and 2999 examples respectively. This shows that BERT-based retrieval models do require a significant amount of supervision before we can deploy them in a real-world setting.

## 7 Conclusion and future work

This paper presents experimental results and analyses on the effectiveness of using recent pre-trained language models to build COVID-19 related QA systems. We evaluate BM25 and unsupervised

BERT-based QA models on a COVID-19 question similarity dataset carefully annotated by public health experts from JHSPH and find that although these perform decently, achieving NDCG@1 of around 0.42-0.52, they are not performing at the level necessary in the real-world environment. When further applying these QA models to an infobot environment, the unsupervised models get poor F1 scores of around 0.35 and it is difficult to find good threshold values that can balance precision and recall. To facilitate future research, we are releasing BERT-NLI[9] and TwitterBERT[10], which were fine-tuned on (input question, candidate question) pairs from our dataset. We are also building a larger COVID-19 question similarity dataset with twenty candidates for every input question. We will publicly release our dataset in the future.

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
