# OpenReview forum: "An Analysis of BERT FAQ Retrieval Models for COVID-19 Infobot"
_aclweb.org/ACL/2020/Workshop/NLP-COVID — Submitted to NLP-COVID-2020_

### Official Review · AnonReviewer3 · 2020-06-09
**Important work but lacks important details**

**Rating:** 4
**Confidence:** 4

**Review:**

QA for COVID-19 is clearly an important topic at this very moment. Authors compile a dataset of questions and related questions and answers and experiment with a number of methods based on similarity of questions to questions and questions to potential answers.

While, I'd see value in this work, but cannot see how this is ready for publication for the following main reasons:
1- writing:
- Reading the abstract, I'm left with what is this work about. What authors have done, what's the contribution?
- It is hard to follow this paper. Many ungrammatical sentences, sentences which have no beginning or end. I'd assume there was rush to get this submitted but can't see this on any acceptable levels of publication.
2- Methods
- what are the similar methods of Q-Q and Q-A out there. Why there isn't any reference in the mini-literature review there?
- How exactly the dataset is complied and annotated is not clear enough. This is the main contribution of this work. Why 0-100 scale is picked and how that is going to be used? How that is helpful in the evaluations? How can you enforce that scale in those metrics picked?
- Might have missed this, but was the dataset going to be made public?
- More statistics on the dataset is needed as what percentage of the questions got a scale that is not zero or 100. The only example in the paper only has 0 and 100.

Overall, while this is a valuable start, it is hardly ready to be published.

---

### Official Review · AnonReviewer1 · 2020-06-11
**Potentially valuable dataset, but requires additional description**

**Rating:** 4
**Confidence:** 4

**Review:**

This work creates a COVID-19 question similarity dataset in consultation with experts from Johns Hopkins Bloomberg School of Public Health (JHSPH), and subsequently evaluates the performance of several BERT-based models on Q-A and Q-Q based tasks.

The dataset itself is perhaps the most valuable and timely contribution. Depending on the questions it contains, it could be valuable to other related efforts like TREC-COVID. It's also fantastic to see a comparison to a baseline like BM25 in addition to the BERT-based methods proposed.

However, several aspects of this paper feel unfinished and it may benefit from the following considerations:

- The description of the dataset leaves many details wanting. Why was a scale of $0 -- 100$ scale used over a Likert-type scale? What's the distribution of similarity scores that appear in the dataset? Why is the dataset going to be released "soon" rather than alongside the paper (a URL could have been provided that was populated upon publication)?

- The writing needs significant edits. Grammatical errors (e.g., subject-verb agreement, commas, footnotes before punctuation, missing citations, etc.) are distracting, but amount to minor disruption. Instead, it's more concerning that the current presentation fails to convey important information. E.g., it's clear from the abstract that a dataset is being constructed, but it's unclear what types of models are being provided as a baseline comparison and what I should take away from the work outside of the dataset.

- The framing could use improvement. While it's not required to enumerate contributions, in this case doing just that would allow readers to quickly understand the take-away messages and allow them to evaluate the remainder of the work with respect to how well it substantiates those claims.

---

### Official Review · AnonReviewer5 · 2020-07-02
**Dataset can be of interest, but creation process is strange**

**Rating:** 4
**Confidence:** 3

**Review:**

This paper presents a pipeline for question-answering for covid-19. They create a novel dataset of covid-19 related questions and answers from multiple websites. Manual annotation of a subset of answers is collected. Various models, including BERT, Roberta, CovidBERT, and TwitterBERT, are evaluated on the dataset in unsupervised and supervised settings. Finally, an infobot is created using the best performing model.

The main novelty of the work is the creation of the dataset, which can be a useful resource for the community, if done correctly.

However, the annotation process seems strange to me. Currently, for every question, only five QA pairs are rated per relevance as retrieved by the BM25-based QA system. However, in creation of a true ground truth of relevance, manual annotators should rank the answers from all the candidate answers, instead of the five returned answers. The approached used is a form of retrieval engine evaluation, instead of ground-truth generation.

Finally, ratings between 0 to 100 is too broad, in my opinion. Why not use a well-defined 0-5 or 0-10 scale?

---

### Decision · Program_Chairs · 2020-07-04

**Decision:**

Reject

**Comment:**

Based on the feedback from the reviewers, we will be unable to proceed with publication of this work.

Thank you for your submission, and we hope that you find the feedback valuable as you further develop this work.